# Cumulative Impact of Testing Factors in Usability Tests for Human-Centered Web Design †

**Alexander V. Yakunin * and Svetlana S. Bodrunova ***

School of Journalism and Mass Communications, St. Petersburg State University, St. Petersburg 199004, Russia

* Correspondence: a.yakunin@spbu.ru (A.V.Y.); s.bodrunova@spbu.ru (S.S.B.)

† This paper is extended from the following conference paper: Yakunin, A.V.; Bodrunova, S.S. *Cumulative Distortions in Usability Testing: Combined Impact of Web Design, Experiment Conditions, and Type of Task Upon User States During Internet Use*; LNCS, Design, User Experience, and Usability: UX Research, Design, and Assessment; Meiselwitz, G., Ed.; In Proceedings of the International Conference on Human—Computer Interaction, HCII 2022 Virtual Event, 26 June–1 July 2022; Springer: Cham, Switzerland; Volume 13321, pp. 526–535.

**Abstract:** The study examines the cumulative impact of factors that affect usability testing for user-centered web design, namely the so-called 'contextual fidelity model' factors that include product properties, task features, user traits, and environment/context factors. Today, the design, user experience and usability (DUXU) research experiences a lack of studies that would assess combinatorial, rather than individual, effects of these factors upon user performance. We address this gap by seeing both independent factors and the resulting user states as complex and dynamic, and testing the combined impact of aesthetic quality of websites, user traits, and individual/group experiment settings upon formation of two dysfunctional user states that critically affect user performance, namely monotony and anxiety. We develop a research design that allows for assessing the combinatorial effects in formation of user dysfunctionality. For that, we conduct a study with 80 assessors of Russian/European and Chinese origin in individual/group setting, employing two types of tasks and websites of high/low aesthetic quality. As the results of our experiment show, group task solving enhances the synchronous impact of website aesthetics and task features upon user states. Interaction of high-quality design, group environment, and monotonous tasks provides for an antagonistic effect when aesthetic layout in a group environment significantly reduces the fatigue rate. Low aesthetic quality in a group environment leads to cumulative enhancing of dysfunctionality for both monotony and anxiety. We conclude by setting questions and prospects for further research.

**Keywords:** user-centered design; web usability; DUXU factors; contextual fidelity model; functional user state; cumulative effects in usability testing; U-index



## 1. Introduction

The paper is dedicated to exploring the combined impact of factors that affect the results of usability testing. Usability testing is a key procedure that allows for making sure that the quality of user interfaces is good enough; moreover, it allows for testing technical, structural, and aesthetic features of interfaces, including the latter for online media.

A substantial number of studies of usability testing orients itself to the factors affecting usability and united under the umbrella of the so-called 'contextual fidelity' model [1–3]. The model includes the four factors such as the test conditions, participants' traits, properties of the tested product, and task features. Together, they form the conditions of usability testing that can significantly affect both the testing accuracy and the final result of the tests. Selection of these factors as the main overarching parameters influencing user experience and efficiency dates back to the 1970s, and they have been investigated in a large number of studies before being aggregated, recently enough, into one model that may be used for assessing their relative importance and cumulative impact upon usability of interfaces.

However, today, most studies assess the influence of these factors, viewing them as separated from each other and not addressing their cumulative impact; this is why the 'contextual fidelity model' as such has not yet been in wide use, while the four factors were acknowledged as contextual and shaping DUXU experiments long ago. In the overwhelming majority of studies of user interfaces, various characteristics within individual factors are investigated [4–6]. Such studies focus on specific aspects of user behavior during the test [7], of testing tasks [8], of the tested product [9], or the test context [10]. Unfortunately, in most cases, they do not attempt to establish relationships between these factors; neither they assess whether the 'contextual fidelity' factors affect each other during test procedures. In particular, testing environment studies look at laboratory vs. field testing efficiency [11,12], or, e.g., the impact of remote testing on the overall user experience [13]. Analogously, product-focused research unidirectionally examines the impact of usability upon objective user's cognitive productivity [14], as well as its influence upon subjective perception of quality, including perceived usability [1] and perceived aesthetics. As a rule, in most of such studies, the results test a direct relationship between individual parameters of subjective user evaluations of their experience and/or objective user performance, on one hand, and the aesthetic quality of web portals, on the other [15,16], without assessment of possible other factors and/or their combinations that may shift the results within the experimental setting.

Similarly, the influence of aesthetic quality of a website upon user experience has become a growing area of research—which, however, remains quite narrow in its scope and highly under-researched. In it, studies of combinations of aesthetic features, task qualities, and experimental conditions are almost non-existent.

In particular, the effects of web aesthetics upon user experience and satisfaction can be divided into subjective and objective ones. Subjective effects are linked to changes in user perception of certain qualities of a product, due to the layout aesthetics. The most significant effects here emerge under the impact of the relationship between aesthetics and perceived usability, when, as experiments show, a product with higher aesthetic quality is also perceived as more convenient to use.

The influence of product aesthetics upon objective criteria of usability and, in particular, upon cognitive performance, remains under-investigated, despite a number of important studies in this area. Here, a certain contradiction in the research results may be spotted. Thus, on one hand, a positive effect of aesthetics upon user motivation is stated, and, in its turn, the increase of this positive effect positively affects the overall performance in solving the task [17]. On the other hand, some researchers have found a direct relationship between aesthetic design and performance degradation. That is, the time to complete a task increases as the aesthetic quality of a product grows [1]. This contradiction is not well-articulated in the current studies, and other factors that may affect the results of experiments are rarely taken into consideration.

The studies of cumulative impact of contextual factors upon user experience are only an emergent area today, which, as we show below, may be divided into two small research zones. Our paper aims at adding to the first line of these studies by testing the cumulative impact of several elements of the 'contextual fidelity model' upon user experience, especially on the formation of dysfunctional user states. We conducted a series of experiments that demonstrate the multi-dimensional effects of aesthetic quality of the product, task change, and differing (individual/group) testing environment dimensions upon the formation of dysfunctionality in user performance. We added to usability studies and development of user-centered design by showing that the experimental settings should be assessed complexly.

We also introduce the theory of (dys)functional user states to the DUXU studies by proving that user experience may lead to formation of dysfunctional states of mind/mood like monotony or anxiety, leading to substantial drops in cognitive efficacy and emotional satisfaction. Just as we see the factors that shape user experience as an inter-related and

dynamic complex, we see dysfunctional user states as complex and dynamic responses by individuals to potentially extreme user experience.

The remainder of the paper is organized as follows. Section 2 goes deeper into the current literature on individual factors of the 'contextual fidelity model.' Section 3 continues the literature review by showing the trends in more complex multi-factor assessment of usability and describes the approach to (dys)functional user states as the complex resulting states during/after task completion. Section 4 poses the research questions and hypotheses. Section 5 focuses on the methodology, including the design of the experiment and its variables; assessor groups construction; the nature of tasks; performance assessment criteria, methods and metrics; and the experimental procedures. Section 6 provides the results of our experiments, and Section 7 discusses them and their limitations. Section 8 concludes the paper and sets prospects for future research.

## 2. Literature Review: Contextual Fidelity Factors and the Lack of Complexity in Today's Research on Them

As already mentioned above, in many cases, product-oriented studies of particular design features are focused on finding patterns between a narrow range of its qualitative characteristics (one or two features of usability and/or aesthetics) and a user's state, where user's state is the target category that is often used as the dependent variable. Obviously, posing research questions this way does not capture the possible multi-dimensional relationship of product properties with circumstances of testing, user traits, and task features.

The problems of research on individual model factors are that:

- most authors focus only on one factor and its impact upon user experience, which makes cumulative impact of several factors slip away from the researchers' attention;
- it is rare that authors use social theories beyond pure perception and/or cognitive studies for assessment of the contextual fidelity factors, while user experience has social aspects and there are theories that may be applied to better understand the user states during task performance;
- the proxies defined by scholars for researching upon contextual fidelity factors are usually very narrow. For example, web design quality is assessed via comparing two types of menus, without all other aspects being taken into account.

Being aware of these limitations, we addressed them in our approach. Thus,

- instead of focusing on one factor, we assess cumulative effects of them all;
- we see these factors, as well as user states, as multi-faceted and complex;
- this is why we apply theories and methods that allow for a significantly more complex assessment of both the contextual fidelity factors and user states.

Below, we review the literature focusing mostly on three research areas. First, we will focus on three factors of the 'contextual fidelity model.' Thus, we will review the works on user traits, especially in their interlinkage with task/product features in relation to aesthetic perception of an interface. In our earlier works, we focused on aesthetic quality of web design in general [18,19] and page navigation elements in particular [20]. Here, we will go further and show how user traits in combination with task/product features affect interface usability seen via user functionality. To this, we will add the research on testing conditions, especially on individual/group testing. Second, we will review the works that aim at studying the cumulative impact of contextual factors upon user experience. Third, we will add a review of works on the user's states as our target category. We will give special attention to the so-called (dys)functional user states and will provide the conceptual background for our study of cumulative impact of contextual factors upon formation of dysfunctional user states during task performance.

All these parts contribute to our research design presented in Section 5.

## 2.1. User Traits in Usability Studies

The research on user traits as an element of the 'contextual fidelity model' and a factor that affects both user performance and perceived design quality, is limited, despite various effects of user traits being actively discussed. Despite the active scholarly interest in this topic, most research focuses on single user traits or experience features and their causal relationship with each other; e.g., the impact of user incompetence on user motivation and self-efficacy [16,21,22] is studied. In particular, within this area, our own research has focused on the relationship between the aesthetic quality of a product, the efficiency of user actions in solving a problem, and perceived usability [14]. However, here, we need to note that this research sub-field is so scarce and non-systemic that, until today, individual user traits have not been systematized, classified, or grouped. Thus, users' inborn physical traits (like gender or age) have not been distinguished from rational mental capacities (like competence, education level, experience etc. [23,24]) or from cognitive and emotional states that have temporal character and may change within or after the experiment, not only as a result of the experiment but also by other reasons [25]. Moreover, in many works, user traits are seen as both independent and target variables, like in the studies of the impact of user incompetence on user motivation and self-efficacy [12,14,17].

As for today, the user trait that provides for the biggest and most complex difference between various groups of assessors is cultural belonging, as it captures many smaller-scale features of visual perception, from, e.g., traditions of color perception to left-right vs. up-down reading. The cross-cultural differences in aesthetic perception within the 'East–West' cultural paradigms have been grounded in many studies (see a review in [26]). According to these studies, the same aesthetic quality of web design may be perceived and assessed differently by representatives of different cultures. This is why we have divided the assessors into two macro-groups, each of them belonged to one cultural community. The East was represented by Chinese students and the West by students from Russia and Europe. In terms of visual design, European and Russian visual cultures are more similar that those of Russia and China. This is why we see such representation as legitimate.

## 2.2. Product Features: Complex Assessment of Web Design Quality

In our earlier studies, we have shown that, today, the quality of web design is usually assessed via proxies that only focus on individual design features, like color, page grids, menu structure, or spacing [12,14]. This does not allow for complex assessment of web design in terms of the relations between web aesthetics and user experience.

Moreover, the results of such studies are also contradictory and non-systemic. They demonstrate both expected patterns (e.g., high aesthetic quality of a product increases perceived usability [13]) and paradoxical dependencies (e.g., high quality design contributes to lower cognitive performance [27,28]). Resolution of the latter paradox may lie in a non-direct relationship between aesthetic quality of layouting and user performance, as this connection may be conditioned by other factors that affect user experience [18,29]. However, at the present stage, there is a significant lack of research on the interaction of various testing factors, despite several important exceptions ([1,30–33]; see below).

On the other hand, most web design schools and their textbooks focus on two measurable levels of web design, which we have called the macro- and micro-levels of page construction. The macro-level deals with the general layout structure of the page, while the micro-level assesses the design quality on the level of syntagma and typography.

We have used this approach in constructing the method for complex assessment of aesthetic features of design in relation to usability. Our approach has been tested and has shown its applicability for various types of web design, including minimalistic designs, non-professional design, and various types of menus [12,14,16]. We have used indexing as an approach and have elaborated the following categories:

- On the macro-level: overall type of layout; layout module structure; vertical spacing; page zonation; creolization of the layout;
- On the micro-level:

    ○    syntagma: line length, line length in title block, leading (inter-lineage spacing);

    ○    typography: contour contrast, tone and color contrast with background, font adaptivity, x-height, font and line length combination.

Each of the 13 chosen parameters were given values (0; 1) or (0; 1; 2). The overall maximum index of a page equaled to 22. Thus, for our current research, we evaluated several web pages and assigned them the U-index means. Then, we selected the pages with high aesthetic quality (U-index > 17) and low aesthetic quality (U-index < 8). On both types of pages, semantically different tasks were performed (see below).

*2.3. The Testing Environment: Individual vs. Group Task Performance*

On earlier stages of usability testing, testing environments were probed for many smaller factors, but these were mostly tested for individual assessors [1,19]. However, in today's research on effects of group interaction upon efficiency of solving non-trivial tasks, the importance of social environment has been demonstrated [16,17]. The approach to project management called collaboration-based problem solving (CPS) has recently attracted a lot of attention from DUXU scholars. CPS is a modern method of resolution of heuristic tasks in creative projecting for media projects, new information technologies, and innovative approaches to education [34]. Within CPS, in-team interaction is crucially important [35].

The two major limitations of the current applications of CPS to usability studies originate in the fact that individual vs. group testing is not combined with task differences and user traits. Moreover, the impact of group environment is only studied within a limited array of tasks, first and foremost the heuristic ones. Additionally, CPS aims at searching upon optimal means of group interaction and not upon cognitive consequences of task-solving. The dependence of user experience upon the nature of tasks is not taken into account, and user states are only fixed for the sake of finding optimal forms of group communication. This is why it is necessary to pay separate attention to group-based testing conditions and reshape the understanding of the role of group testing in accumulation of effects in usability tests.

## 3. Literature Review: Towards the Assessment of Cumulative Impact of Contextual Fidelity Factors and Complexity of User States in Usability Testing

*3.1. User Traits vs. Task Features/Product Features, and Their Crossroads in Aesthetic Perception*

In studies dedicated to task features, most often, it is the quantitative characteristics of tasks that become the research focus, e.g., when solving single and dual tasks is compared [1,14]. At the same time, a number of studies deal with the issues of the influence of task properties upon the manifestations of experience dependent on user traits, where user traits are not only sociodemographic (like age or gender) but also cognitive. These are, i.e., the effects of the influence of the cognitive complexity of the task upon aesthetic perception [36,37]. Such studies link the process of aesthetic perception and judgment to cognitive activity of a person, with his/her ability to judge reality in particular ways. Aesthetic ability, in this case, is a complex of interrelated phenomena of mental life, including memory, experience, knowledge, insight, imagination, judgment, and creative activity [38]. Along with imagination, aesthetic perception is included in the general complex of human cognitive activity. Thus, it may depend on both cognitive ability and mental/cognitive experience of a given user. In particular, the authors [39] pointed out to the connection between aesthetic evaluation and the visual complexity of the task. The dependence of aesthetic preferences upon the degree of familiarity with the product was also documented [40]. In another study, the dependence of aesthetic evaluation upon both aesthetic context and the respective cognitive experience was demonstrated [41].

In this regard, of particular interest are the studies that convincingly show that differences in aesthetic perception are largely determined by differences in the semantic complexity of assessed objects [42]. In such studies, it becomes possible to establish proven links between the cognitive complexity of the task or the tested product, on one hand, and

users' aesthetic judgment, on the other. Another study [41] of the impact of semantic identification of elements of graphic interfaces upon their aesthetic assessment has demonstrated the dependency of aesthetic judgment on design quality upon visual/textual variability of the task (in particular, the differences in judgment upon pictograms, ideograms, and textual links). Interestingly, according to this paper, it is task variability, not the level of cognitive load in the task, that casts impact upon aesthetic judgment.

While recognizing the importance of such relational studies, we also cannot help stating that most of them are limited to assessing quantitative differences in tasks, in particular the varying cognitive complexity and/or users' cognitive load within same-type tasks. At the same time, semantic task features and their connection with the testing environment remain understudied. In general, we see that the studies of perception of interfaces are today trying to become more complex and include more factors of task/product provenance into the models assessing the impact of user traits on aesthetic judgment upon interfaces. Still, at the present stage, there is lack of research that would assess the effects of the interaction of various testing factors, especially the cognitive user traits and the test environment features.

This is why we have oriented our study to creating tasks that would correspond to the idea of cumulative effects in user experience. The tasks that we have created for our experiment differ from other experiments in the DUXU area in three ways. First, they are semantically divergent, as they are oriented to inducing two different complex user states that emerge during the experiment (see below). Second, they allow for detecting the differences in use of aesthetic/non-aesthetic web portals, as assessed by the U-index (see above). Third, they are suitable for both individual and group performance (see below). The tasks themselves are described in the Methods section.

### 3.2. The Research on Cumulative Nature of User Experience

Recently, a research area dedicated to studying the cumulative effects in user experience has started to form [43]. This approach sees user experience not as objective in nature and shaped by factors external to the assessor but as a continuous, dynamic, and complex process guided and shaped by user activity [4,5]. The approach develops mostly via longitudinal studies that compare the quality of user experience on different stages of product use [11,44–46].

The methodology of such studies is based on the premises of activity theory [47]. It sees problem-solving as a process of qualitative transformation of the subject of activity by building skills and competence over time [47,48]. According to this theory, users' practical activity has a hierarchical structure, at the top of which is the desire to improve functional capabilities of the subject of activity, and at the basis lie specific operations and algorithms designed to solve a pragmatic problem [49]. The process of a user's interaction with the product, therefore, implies the interdependence of the subject and the object where user experience is formed both *by* solving the problem and *in the process* of solving it [36,50,51]. In the context of human-computer interaction, this implies a close relationship between the factors of the relational context, as well as between these factors (like the nature of the task), on one hand, and the user state, on the other, especially on the motivational level [52]. Activities and users thus influence each other; therefore, proponents of this approach describe user experience as subject-object interaction [53,54].

The merit of such studies is that they represent user experience as an integral and dynamic complex that can change over time under the influence of interaction factors, in particular the context of the activity and task features. However, one cannot help noting that, in the abovementioned studies, attention is predominantly paid to activity fundamentals, while specific effects that arise from interaction of its key factors—the research context, task features, product properties, and user traits—remain heavily understudied.

### 3.3. User Functionality as the Target Category in User Experience Studies: (Dys)functional User States

If the 'contextual fidelity model' describes the four major independent factors that affect user experience, and the activity theory sees their interaction as crucial, the dependent (or target) variables in the DUXU studies describe efficiency evaluation, both objective (user performance in objective measurements [25,55]) and subjective (users' levels of satisfaction, pleasure, and/or perceived efficiency [56–58]). These two aspects do not always correspond to each other and need to be measured in combinations in order to assess the true quality of user experience. However, if the independent factors that shape user experience are today more and more seen as a dynamic complex, the same should go for the results of their influence. The resulting states of a user's mind and mood, both during and after performance, need to be seen as complex and dynamic just as well. Moreover, as mentioned above, the 'positive' qualities of product (say, 'beautiful' design) may negatively affect user performance and its subjective evaluation, especially when combined with other factors of testing, and the resulting user state may linearly depend on none of individual impact factors. However, the DUXU studies do not yet share any universally agreed-upon conceptualizations for dynamic and complex user experience.

In our previous works, we have come to a conclusion that a suitable concept of how the surrounding factors affect labor performance may be found in psychology of work and professions. As shown in [18], critical changes in users' mental and emotional states while performing complex tasks are best described by functional states theory. This branch of psychology of professions deals precisely with how job performance forms complex states of mind/mood that, in their turn, critically affect the performance quality and, thus, need to be known and, in some cases, struggled with. The Russian psychological school has produced several important works in this respect, which we will mainly use as the basis of elaborating upon the target variables of our study. In them, the *functional state* of a worker (of a user/assessor, in our case) is an integral and simultaneous complex of mental and bodily functions that determine professional performance [12]. As user states are assessed in terms of both working efficiency, on one hand, and the state of individual mind, mood, and behavior, on the other, the scholars insist such user states may be either functional (that is, raising objective performance and subjective satisfaction) or dysfunctional (that is, lowering performance and subjective satisfaction).

Any (dys)functional state arises in the process of performing a certain activity and is the result of the interplay of psychophysiological factors. (Dys)functional states show up not in changes in individual psychophysiological manifestations (attention, performance, etc.), but in the dynamics of a whole complex of such indicators, forming stable, recognizable trends called patterns [21,59]. Just like the activity theory, the theory of functional states sees the user states both as a background against which mental processes unfold and activities are carried out, and as the resulting behavioral dynamics. On one hand, real changes in the structure and content of activity are associated with the functional state. On the other hand, the very structure and content of the activity form the functional state and provide opportunities for a holistic study of all its components [60]. Such a dialectic of activity and experience best suits the task of describing the results of the interaction of activity factors ('environment + product properties') and UX factors ('psycho-physiological state of the user + cognitive structure of the task').

(Dys)functional user states comprise cognitive and emotional states that combine and altogether result in definite, specific, and detectable task-induced behavioral changes. User states, thus, may foster or diminish functional productivity of work. Dysfunctional user states that form due to unfavorable combinations of task performance factors deserve special attention, as they may significantly diminish the efficacy of work performance; we hypothesize that they affect performance on online tasks just as well. According to Leonova [61], the wide range of possible emotional-behavioral reactions and modes of activity may be reduced to several distinct dysfunctional states that vary in their destructive effect upon working performance, from desire to activity shifts to exhaustion to panic to

extreme distress. However, the extreme states are very rare for Internet use; much more often, boredom, tiredness of monotonous work, and anxiety are experienced [62].

We have selected two dysfunctional states, *monotony* and *anxiety*, as target states, as they are: (1) relevant for Internet use; (2) opposing each other in terms of emotional arousal and types of performance errors that help detect them; but (3) equally damaging in terms of decrease of both objective performance quality and subjective pleasure of interface use.

Experiencing these states manifests via the two respective stable sets of features (patterns) that may determine the specifics of user experience. Being detectable, they allow for accurately identifying a dysfunctional state that a user encounters, with the help of cognitive and emotional markers that may allow for the use of quantitative methodologies. In particular, in the dysfunctional state of anxiety, a drop in intellectual lability is accompanied by a violation of concentration due to stressful conditions for performing a task. Additionally, this state is characterized by an acceleration in the pace of work with a simultaneous increase in errors and the resulting tendency towards an increase in unproductive information processing strategies. The ability to switch attention between different activities also tends to increase but is characterized by hypersensitivity to distractions and poor self-control. At the same time, the degree of awareness of one's own actions decreases, and emotional stress grows, reaching the level of distress in the extreme values [30,60]. In the dysfunctional state of monotony, there is also a drop in intellectual lability, but, in this case, it is due to other reasons. First of all, narrowing of the amount of attention and the decrease in the ability to switch between different tasks, due to increased fatigue, are responsible for the formation of monotony. The overall pace of activity in solving the problem slows down (in particular, making choice is complicated), the efficiency of switching from one type of reaction to another decreases. The level of emotional stress, as a rule, falls, while apathy and fatigue at their extreme is approached by the users [63].

Given all the aforementioned, our approach is based on treating both the process of task performance (as shaped by the contextual fidelity factors) and the resulting user response (in the form of a (dys)functional user state) as complex multi-faceted processes that form in time. More precisely, we tested the cumulative impact of contextual fidelity factors by assessing user dysfunctionality of two types. For this, we state several research questions and hypotheses.

## 4. The Research Hypotheses

Based on what is said above, we have developed an experiment-based method capable of demonstrating combinational effects in the synchronous interaction of various usability testing factors. The proposed study examines the simultaneous influence of the four testing factors (the individual/group environment of the experiment, the type of task, the product properties, and cultural belonging) upon formation of (dys)functional user states that critically affect the assessors' performance.

In accordance with the conclusions of our earlier works and the above review of literature, we have formulated three research questions:

**RQ1:** How does the combination of task features and web aesthetics affect user performance in terms of formation of anxiety and monotony?

**RQ2:** How does the group format of task-solving affect the interaction of factors that form the user states?

**RQ3:** What are the cumulative effects of the combination of task features, web aesthetics, and testing circumstances?

The RQs were also complemented by three hypotheses, with the regard of which we have then developed the design of the experiment. Our hypotheses reshape the RQs, rather than open them up in a more conventional way. In particular, our RQs are focused on interaction of individual contextual fidelity factors (e.g., web aesthetics vs. the testing format), to make the results more comprehensible being narrated bit by bit. The hypotheses, in turn, take into consideration the previous findings by other authors and hypothesize the effects created by many factors simultaneously. RQ1 and RQ2 both contribute to H2 and

H3, while RQ3 relates more to H1. We realize that this approach is unusual but hope this order will help better convey the received results.

**H1:** In the conditions of group problem-solving, the effects that arise due to the simultaneous influence of the website aesthetics and the task features upon the user states will be cumulative, i.e., will intensify.

**H2:** The interaction of the high aesthetic quality of the website and tasks of both types (inducing monotony and anxiety) in the conditions of group testing will lead to a decrease in cognitive efficiency and an increase in emotional stress (anxiety), also in comparison to individual testing.

**H3:** The interaction of the low aesthetic quality of the website and tasks of both types under conditions of group testing leads to unfavorable emotional states corresponding to the nature of the tasks (an increase in anxiety for a forced task and fatigue for a monotonous one), while cognitive efficiency decreases in both cases, also in comparison with individual task-solving.

## 5. Methods and Assessor Groups

### 5.1. The Research Design

We have elaborated the experimental research design based on the assumptions stated above. Our variables were constructed the following way:

- Contextual fidelity factors (independent variables):

    (1)　product features $\geq$ web aesthetics quality $\geq$ U-index (13 parameters);
    (2)　task complexity $\geq$ monotony-inducing and anxiety-inducing tasks especially elaborated for the experiment;
    (3)　user traits $\geq$ cultural belonging of respondents, as cultural belonging implies many individual factors of visual perception, from traditions of color perception to left–right vs. up–down reading;
    (4)　experiment conditions $\geq$ group vs. individual performance, as this division, according to previous research, may affect the results to the biggest extent.

- User states (dependent variables): The states of monotony and anxiety as complex dysfunctional states, instead of measuring particular eye reactions or time of performance. Measured by both rational and emotional parameters, that is, intellectual lability and emotional stress (with four scales for measuring individual emotions).

We see this research design as inclusive and strict at the same time.

### 5.2. The Assessor Groups

Initially, 80 assessors comprised two macro-groups based on cultural belonging. Both macro-groups performed the test tasks in two groups that differed by participation formats, namely the individual and group participation. The difference between group and individual testing formats lies in the presence of other people immediately around the test-taker during task completion. In the group test format, all participants in the experiment perform the task on the same computer one after another, while they see and hear each other. In the individual test format, each participant performs the test alone and acts according to the instructions that (s)he had received in advance.

Moreover, within each format-based group, there were sub-groups divided according to the task types, namely the monotony- and anxiety-inducing tasks.

For each task, the sub-groups worked with webpages of either high or low aesthetic quality. The quality was measured based on the U-index that we had elaborated in our earlier research (see above and in [18]).

Thus, the research design may be described by the formula '2 (2 × 2) × 2 (2 × 2) + 2 (2 × 2) × 2 (2 × 2)' (see Table 1). This formula, in effect, means that we have four factors to test, with *each* factor having two major sub-groups which, in their turn, divide into smaller groups by each other factor. For 'monotony/anxiety', 'individual/group testing', 'East/West', and 'aesthetic/non-aesthetic design', there are eight groups per each of the two

categories. However, each assessor only has four features to test. For example, sub-group 1 belongs to individual testing + monotony task + aesthetic design + the 'Eastern' culture.

**Table 1.** The research design and assessor groups, 5 people per group.

| Test Format | Dysfunctional State | 'East' | | 'West' | |
|---|---|---|---|---|---|
| | | Aesthetic Design | Non-Aesthetic Design | Aesthetic Design | Non-Aesthetic Design |
| Individual test | Monotony | sub-group 1 | sub-group 5 | sub-group 9 | sub-group 13 |
| | Anxiety | sub-group 2 | sub-group 6 | sub-group 10 | sub-group 14 |
| Group test | Monotony | sub-group 3 | sub-group 7 | sub-group 11 | sub-group 15 |
| | Anxiety | sub-group 4 | sub-group 8 | sub-group 12 | sub-group 16 |

According to the research design, the study sample consisted of 80 students aged 22–23 with at least 9 years of experience of the Internet use. Each group contained 5 people; one assessor could belong to one group only. However, with our research design, each 'contextual fidelity model' factor had 40 vs. 40 students, which is more than in most DUXU studies. One half (40 people) were represented by students from China, the other half by students from Russia and Europe (both groups spoke English at the same level); 40 students underwent individual testing and 40 other students experienced group testing; 40 students performed monotonous tasks, and 40 were subjected to anxiety-inducing tasks; 40 saw aesthetic design and 40 saw non-aesthetic design.

This research design, however, implies that sub-groups are too small for statistical analysis of practically any type. This is why, for our data analysis (see 5.5), we used enlarged groups (10 people per group) which implied that we disregarded one of the 'contextual fidelity model' factors in each case. For example, to test cultural differences between Chinese and Russian/European students, we have conducted two Mann–Whitney tests, one for the differences between aesthetic and non-aesthetic design (regardless of the nature of individual/group testing), and the other for individual/group work (disregarding the differences between aesthetic/non-aesthetic design). This has allowed to assess the cultural differences multi-dimensionally, taking into account the difference in tasks.

After the tests on cultural differences, we merged the East/West macro-groups, thus also enlarging each micro-group to 10 students in the final research design. This has allowed us to calculate the standard deviations for the resulting values in sub-groups with better probability of their significance (see below) and, again, to apply the Mann-Whitney U-criterion test, also with higher validity.

*5.3. Elaboration of Test Tasks*

Our test uses two types of tasks. Their differences are determined by how they influence the user functional state. They are a forced task, which contributes to the formation of the state of anxiety, and a monotonous one, which evokes the growth of monotony/fatigue.

The main parameter that leads to the dysfunctional state of anxiety is the lack of time and information in solving the problem. To form it, it is enough to provide unforeseen changes in the test conditions, e.g., to dramatically complicate the task without prior warning, or to suddenly change the content of the task while reducing the time for task-solving.

In accordance with this, the structure of the forced task that was to foster anxiety was developed: Each user was asked to find a given (actually non-existent) piece of content on a website of the corresponding aesthetic quality within 10 min. Three minutes after the start of the search, the curator of the experiment announced a change in the purpose of the search and a reduction in time to 5 min, then, another minute later, to 2 min.

For formation of monotony, activity with a large number of simple and monotonous movements with very little creativity is crucial. To create the appropriate conditions, the assessors were offered a monotonous task. Within 20 min, they had to prepare a full-text version of the site of given aesthetic quality (in *.doc format, via copy-pasting and correcting

small fallacies that emerge from copying). The exact time of completion of the task was not told to the assessors. The main factors in the formation of monotony, in this case, were the uncertainty on task timing and a significant number of repetitive operations necessary for copying large amounts of web content into a Word document.

As our assessors had to be put into equal conditions, we used the web portals in English which all the assessors had good command of but were not native speakers of. We have used the existing English-language (American) university portals (the Syracuse State University as the high-aesthetic web product and Iowa State University as the low-aesthetic web product; see Appendix A, Figures A1 and A2, respectively) for task creation and testing.

*5.4. Measurement and Scales*

As we have already described above, the differences in product features were represented by two university website layouts with the corresponding high and low aesthetic quality, as measured by the website aesthetic quality index (U-index) [12]. The university websites were pre-assessed and assigned the U-index values.

To measure user performance, we employed two groups of metrics that focused on cognitive and emotional aspects of the user states, respectively.

For cognitive performance, *intellectual lability* was chosen as an indicator. Cognitive performance is an aspect of cognitive load testing that focuses on positive achievement of the assessor, evaluating, e.g., the number of correctly fulfilled tasks, as well as the number of mistakes and performance time. Intellectual lability, in its turn, describes the ability to work efficiently in the multitasking mode, switching from one task to another, as well as the ability to concentrate for a long time. The better the lability, the faster the speed and quality of the task completion [64]. Intellectual lability has been mostly researched upon in the 'practical thinking' studies within education research [65–67], especially in the Russian-speaking studies, while Western studies focused more on emotional lability. However, collaborations between those have brought to life testing methods that may be successfully applied to today's DUXU studies.

Using one of these methods, we have measured intellectual lability using the test for switching attention known as the 'Schulte–Gorbov table' [68], elaborated by the renowned German psychologist Walter Schulte and further developed by founders of the Soviet medical psychology, like Konstantin Platonov and Fyodor Gorbov. We applied it before and after the experiment. The method implies that the participants pronounce figures from 1 to 25 (shown on grey squares of the table where grey and red squares are mixed) and from 24 to 1 (shown on red squares of the table), switching from grey to red and back. The time of task solution and the number of mistakes are both fixed and then are rendered into two respective scales proposed by Schulte/Gorbov. The overall lability is measured for each participant individually and calculated as $A = T - C$, where T is the scaled metric for the longevity of the test, and C is the scaled metric for the number of errors. The smaller A, the worse the performance. We have calculated A for each participant before and after the main task completion, thus detecting the drops and rises (if any) of intellectual lability. Then, we calculated $A_{mean}$ for all the assessors in each sub-group (see below).

*Emotional stress* is the second measure of performance quality that we have applied. It refers to the level of overall stress that an assessor experiences during the task completion. However, the nature of stress is complicated, and this is why it is measured via four sub-parameters, which are called the indicators of self-esteem: In accordance with the method of self-assessment of emotional states proposed in [69], we measured, by the 1–10 Likert scales, the level of:

– anxiety (the 'calmness/anxiety' scale);
– fatigue (the 'energy/fatigue' scale);
– arousal (the 'excitement/depression' scale);
– confidence (the 'self-confidence/helplessness' scale).

The scales, however, are not formed via assessors' putting marks from 1 to 10; instead, the scales consist of statements that describe emotional states and are assigned 1 to 10. This is why the scales might be imperfect and provide for answers more similar than expected (see 7.2); however, this method is more elaborated than simpler Likert scales and reflects the nature of our tasks. The four scales are, somewhat counter-intuitively, structured in a way that higher values lead to diminishment of emotional stress: That is, from 1 to 10, calmness, energy, excitement, and self-confidence grow, while anxiety, fatigue, depression, and helplessness fade. Thus, moving from 7 to 5.5 (the −1.5 drop of good state) means growth of emotional stress.

### 5.5. Data Analysis

For each participant, we have received 10 values, namely $A_{mean}$ and the four Likert-based metrics before and after the test.

First, the results that we received from the assessors were checked for macro-group differences potentially conditioned by ethno-cultural belonging. We wanted to know whether cultural differences cast impact upon the assessors' results, because, if not, they could be disregarded in the further course of data analysis. To test this, we used the Mann–Whitney U-criterion metric. This metric is especially created for comparing means of two small samples of nonparametric data. We have selected this particular metric to test the divergence of culturally-bound user experience, as our groups were not large, the data dispersion was not normal, and, in each case, only two groups, not three or more, were to be compared. Due to small enough sub-groups (which makes the Mann–Whitney test de-facto applicable instead of the Wilcoxon test), we have applied this metric both in the case when dispersions were equal but unknown (for intellectual lability, where the potential dispersion was [0; ∞)) and in the cases where dispersions were equal and known (for emotional stress indicators where potential dispersion was [1,11]).

However, our groups of five participants each were too small to statistically measure effects, even with the metrics especially for small samples. This is why, to test cultural differences before other 'contextual fidelity' factors, we have decided to conduct two statistical tests in bigger groups: one would test the 'East/West' differences with differing tasks and differing aesthetic quality but disregard individual/group differences, and the other would test the 'East/West' differences in individual/group environment disregarding the aesthetic quality of web portals. By doing so, we have received two times larger sub-groups (10 assessors per sub-sample) and conducted two tests to detect the cultural divergence. For the results, see Table 2.

The Mann–Whitney test has shown that cultural differences were insignificant in all cases (even if in three cases, all relating to growth of depression, they came quite close to being significant). This has allowed us to enlarge the sub-groups disregarding the cultural differences, thus creating sub-groups of 10 assessors each.

Second, being interested in the *difference* between the results before and after the test for each participant and each metric, we have calculated *delta* (Δ) for each before/after pair of results, thus measuring the size and direction of effect for each of the five metrics for each participant. Then, we have calculated the standard deviations for 'before', 'after', and delta values in each sub-group, detecting whether the shift between before and after the tests has been significant. We, though, need to note here that standard deviations are not generally used in in the DUXU research, as small number of assessors do not imply that standard deviations are commonly used, and this situation is accepted by the scholarly community (see, e.g., [70–72]). We are using standard deviations to additionally support our findings only, but will anyway orient to metrics meanings themselves as well as to their standard deviations, to show the trends in user objective and subjective behavior before vs. after the task completion.

Third, we have applied the same Mann–Whitney U-criterion test to detect the differences induced by the two factors, namely the aesthetic/non-aesthetic design and the individual/ group environment, for the two types of tasks. The combination of statistical

criteria (standard deviations + Mann–Whitney U-criterion) has allowed us to evaluate the effect (that is, significance of the shift between 'before' and 'after' values) for each metric and to trace the impact of each 'contextual fidelity' factor (e.g., individual/group environment) all other metrics being equal (see Tables 3–6). Again, our groups were small enough to put the major attention to the significance of comparing means, just as to the standard deviations; this is why, in describing the trends in the data, we will be paying attention to both the mean values *and* to their validity/significance intervals.

Our results are presented below.

## 6. Results

### 6.1. Testing the Cultural Differences

As stated above, on the first stage, the results that we have received from the assessors have been checked for macro-group differences potentially conditioned by user traits—that is, by their ethno-cultural belonging. To test this, we calculated the deltas for the values received before and after test for the enlarged groups, as described in Section 5.5 (see Table 2), and compared the deltas' means with the help of Mann–Whitney U-test.

**Table 2.** *p*-values of the Mann-Whitney U-criterion for the 'East' vs. 'West' macro-groups, aesthetic vs. non-aesthetic design and individual vs. group work.

| User States | Metrics | | Aesthetic Design | Non-Aesthetic Design | Individual Work | Group Work |
|---|---|---|---|---|---|---|
| Monotony | Intellectual lability | | 0.739 | 0.315 | 0.218 | 0.912 |
| | Emotional stress | Calmness/anxiety | 0.353 | 0.796 | 0.280 | 0.971 |
| | | Energy/fatigue | 0.481 | 0.684 | 0.912 | 0.912 |
| | | Excitement/depression | 0.052 [†] | 0.218 | 0.247 | 0.055 [†] |
| | | Self-conf./helplessness | 0.853 | 1.000 | 0.280 | 0.315 |
| Anxiety | Intellectual lability | | 0.853 | 0.579 | 0.912 | 0.853 |
| | Emotional stress | Calmness/anxiety | 0.353 | 0.28 | 0.739 | 1.000 |
| | | Energy/fatigue | 0.218 | 0.436 | 0.739 | 0.912 |
| | | Excitement/depression | 0.052 [†] | 0.529 | 0.529 | 0.529 |
| | | Self-conf./helplessness | 0.796 | 0.315 | 0.123 | 0.796 |

[†] the *p*-value comes close to 0.05.

As the results in Table 2 suggest, the results for 'East' and 'West' groups do not differ substantially. In no cases, did the *p*-value in measuring the differences between groups reach 0.05. This allows for confirming the null hypothesis ('the macro-groups do not differ') and conclude on the absence of the impact of ethno-cultural differences in user traits for our research in terms of cognitive and emotional consequences of the experiment. However, in three cases all related to growth of depression, the cultural differences come close to the generally accepted confidence interval (0.05). This may deserve further studies, as this difference is detected for both monotonous and anxiety-inducing tasks performed upon high-quality design.

The absence of substantial differences on the cultural level allows for uniting the 'East' and 'West' datasets. The aggregated results allow for addressing the research questions and hypotheses.

### 6.2. The Results of the Experiment and Their Interpretation

The generalized results are presented in Table 3 (aesthetic vs. non-aesthetic design, individual testing for monotony and anxiety) and 4 (aesthetic vs. non-aesthetic design, group testing for monotony and anxiety) which describe the main outcomes of our experiment, significant results bold. Tables 5 and 6 restructure the findings and show Mann-Whitney results for individual vs. group testing.

*RQ1: How does the combination of **task features** and **web aesthetics** affect user performance in terms of formation of anxiety and monotony?*

Our results show that there are several levels on which both the effects and differences show up. Thus, when aesthetic/non-aesthetic design is juxtaposed to monotony- vs.

anxiety-inducing tasks, we see that our tasks, indeed, have in general worked as inductors of particular target emotions, as well as decreased the assessors' intellectual lability. In particular, for monotony, intellectual lability drops, and fatigue rises, while anxiety remains stable; for the state of anxiety, intellectual lability drops, and anxiety (as emotion) rises, while fatigue grows only slightly and insignificantly. In both cases, excitement may be called stable, even if, for monotony, it slightly decreases, and for anxiety, it slightly increases, which also corresponds to the nature of the tasks.

However, web aesthetics (that is, efficient user-friendly design as measured by U-index) demonstrates divergent patterns in how it affects formation of two different user states. Thus, for monotony, low-quality design definitely decreases intellectual lability for both individual and group testing: $-7.83$ (SD = 3.019) and $-10.98$ (SD = 5.622), respectively, while high-quality design helps diminish the drop of lability in both cases. The drop is significant but small for individual testing: $-4.12$ (1.504), and in group testing on quality web pages, intellectual lability is maintained: $-1.76$ (6.401). For anxiety, though, the effect is reverse. Here, paradoxically but expected due to earlier research, non-efficient design better preserves intellectual lability. For both individual and group environment, the drop of intellectual lability is much weaker for non-aesthetic design: $-16.02$ (2.791) vs. 0.1 (4.423) and $-27.76$ (5.977) vs. $-13.05$ (3.497). For individual testing, it even stays nearly unchanged.

Emotional stress, as noted above, shows up in the key task-corresponding emotions: for monotony, fatigue grows, and for anxiety, emotional anxiety rises, while other emotions remain more or less stable. However, here, we also see the impact of web design quality. Thus, for monotony, a combined effect of aesthetic/non-aesthetic design and individual/group testing regime shows up. In individual testing, fatigue grows equally for both types of design (|1.4|), but, for non-aesthetic design, SDs are higher, thus the nature of change is a bit different: With non-aesthetic design, assessors differ more in whether they got tired or not. For group testing, though, aesthetic design evidently shows non-fatigue (|0.1| (0.568)), while, for non-aesthetic design, the growth of fatigue is even bigger than for individual testing (|1.6| (1.579)). Thus, a cumulative effect of group performance and efficient design that, together, better save the users from fatigue may be stated.

The results for the confidence scale are insignificant in terms of standard deviations, but, as explained above, we still may look at the trends we see in the data. The results show that the aesthetic designs in both cases have slightly increased the assessors' confidence, while the non-aesthetic ones have slightly decreased it, which needs to be studied further on larger samples.

All in all, we may detect the two following groups of effects in combination of task features (the tasks inducing the two dysfunctional states) and product features (web aesthetics as measured by the U-index):

(1) Monotony-related effects:

- Induction of monotony (drop of intellectual lability + growth of fatigue) in individual testing, with slight difference in fatigue dispersion between aesthetic and non-aesthetic design;
- Induction of monotony (drop of intellectual lability + growth of fatigue) for group performance on non-aesthetic design;
- Preservation of user state in monotony-inducing tasks in group performance, if mediated by aesthetic design;

(2) Anxiety-related effects:

- Induction of the state of anxiety (drop of intellectual lability + growth of anxiety) in group testing on both design types and in individual testing mediated by efficient design;
- Preservation of intellectual lability in group testing with non-aesthetic design.

**Table 3.** The results of the experiment, individual testing, aesthetic vs. non-aesthetic design.

| User State | | Metrics | Individual Test | | | | | | Mann-Whitney, Δ *p*-Value |
|---|---|---|---|---|---|---|---|---|---|
| | | | Aesthetic Design | | | Non-Aesthetic Design | | | |
| | | | before Task | after Task | Delta (Δ) | before Task | after Task | Delta (Δ) | |
| Monotony | | Intellectual lability (A$_{mean}$) | **187.05** **(1.057)** | **182.93** **(1.128)** | **−4.12** **(1.504)** | **186.74** **(1.266)** | **178.91** **(2.166)** | **−7.83** **(3.019)** | **0.004** |
| | Emotional stress (10 to 1) | Calmness/anxiety | 7 (0.471) | 7 (0.480) | 0 (0.667) | 7 (1.490) | 6.9 (1.449) | −0.1 (1.853) | 0.912 |
| | | Energy/fatigue | **6.9** **(1.197)** | **5.5** **(0.527)** | **−1.4** **(1.505)** | 6.9 (1.449) | 5.5 (1.649) | −1.4 (2.547) | 1.000 |
| | | Excitement/depression | 6.9 (0.567) | 6.9 (0.316) | 0 (0.817) | 7.1 (1.728) | 7 (1.247) | −0.1 (0.876) | 0.684 |
| | | Self-conf./helplessness | 6.85 (1.475) | 7.1 (1.100) | 0.3 (1.767) | 7.1 (1.370) | 7 (0.816) | −0.1 (0.876) | 0.353 |
| Anxiety | | Intellectual lability (A$_{mean}$) | **187.15** **(1.285)** | **171.13** **(3.067)** | **−16.02** **(2.791)** | 186.94 (0.751) | 187.04 (4.247) | 0.1 (4.423) | **0.000** |
| | Emotional stress (10 to 1) | Calmness/anxiety | **7.1** **(0.875)** | **6.1** **(0.800)** | **−1** **(0.471)** | 6.8 (1.135) | 5.7 (0.823) | −1.1 (0.568) | **0.000** |
| | | Energy/fatigue | 6 (0.670) | 5.9 (0.737) | −0.1 (0.568) | 6.1 (0.567) | 5.9 (0.737) | −0.2 (0.632) | 0.739 |
| | | Excitement/depression | 6.9 (0.567) | 7 (0.471) | 0.1 (0.567) | 7.1 (0.994) | 7.2 (1.135) | 0.1 (0.568) | 1.000 |
| | | Self-conf./helplessness | 7.1 (1.370) | 7.3 (1.251) | 0.2 (0.789) | 7 (1.247) | 6.9 (0.875) | −0.1 (0.994) | 0.436 |

*Note.* The values with standard deviations lower than deltas and *p*-values $\leq 0.05$ are highlighted.

**Table 4.** The results of the experiment, group testing, aesthetic vs. non-aesthetic design.

| User State | | Metrics | Group Test | | | | | | Mann-Whitney, Δ *p*-Value |
|---|---|---|---|---|---|---|---|---|---|
| | | | Aesthetic Design | | | Non-Aesthetic Design | | | |
| | | | before Task | after Task | Delta (Δ) | before Task | after Task | Delta (Δ) | |
| Monotony | | Intellectual lability (A$_{mean}$) | 186.9 (3.478) | 185.14 (8.230) | −1.76 (6.401) | **186.92** **(3.030)** | **175.94** **(3.815)** | **−10.98** **(5.622)** | **0.003** |
| | Emotional stress (10 to 1) | Calmness/anxiety | 6.9 (1.286) | 6.8 (1.135) | −0.1 (0.316) | 7.1 (0.875) | 6.9 (1.286) | −0.2 (0.919) | 0.436 |
| | | Energy/fatigue | 5.9 (0.737) | 5.8 (0.788) | −0.1 (0.568) | **6.9** **(1.197)** | **5.3** **(1.059)** | **−1.6** **(1.579)** | **0.015** |
| | | Excitement/depression | 7.2 (0.632) | 6.9 (0.567) | −0.3 (0.483) | 7.1 (1.286) | 6.8 (0.918) | −0.3 (0.949) | 0.684 |
| | | Self-conf./helplessness | 6.5 (1.080) | 6.8 (0.918) | 0.3 (1.059) | 6.7 (2.110) | 6.9 (2.020) | 0.2 (0.632) | 0.853 |
| Anxiety | | Intellectual lability (A$_{mean}$) | **186.96** **(1.028)** | **159.2** **(5.860)** | **−27.76** **(5.977)** | **186.99** **(0.741)** | **173.94** **(3.048)** | **−13.05** **(3.497)** | **0.000** |
| | Emotional stress (10 to 1) | Calmness/anxiety | **6.7** **(1.330)** | **5.1** **(1.286)** | **−1.6** **(0.699)** | 7.1 (0.875) | 5.7 (0.823) | −1.4 (0.966) | 0.529 |
| | | Energy/fatigue | 5.9 (0.737) | 5.8 (0.788) | −0.1 (0.568) | 6.1 (0.875) | 5.9 (1.449) | −0.2 (1.549) | 0.853 |
| | | Excitement/depression | 6.9 (0.567) | 7 (0.471) | 0.1 (0.568) | 6.9 (0.737) | 7 (0.816) | 0.1 (0.316) | 0.971 |
| | | Self-conf./helplessness | 6.9 (0.500) | 7 (1.054) | 0.1 (0.738) | 6.1 (1.663) | 5.9 (1.286) | −0.2 (0.422) | 0.393 |

*Note.* The values with standard deviations lower than deltas and *p*-values $\leq 0.05$ are highlighted.

Summing up, in our data, efficient design preserves assessors who perform monotonous tasks in groups from lability drops and growth of fatigue. At the same time, 'bad' design seems to prevent decrease of multi-tasking in individual performance upon anxiety-inducing tasks, which does not seem logical but corresponds to previous findings [17]. Even if the pattern for anxiety-inducing tasks slightly favors non-aesthetic design and demands more studies, we still recommend following the recommendations previously set for high-quality design in our work [18], as monotonous tasks are met much more often in online pastimes, and struggling with monotony is in demand among many wide groups of online users who face the circumstances of group work, including office workers and students.

*RQ2: How does the **group format** of tasks solving affect the interaction of factors that form the user states?*

Tables 5 and 6 reproduce the results of the experiment for clearer juxtaposition of individual vs. group testing results for aesthetic (Table 5) and non-aesthetic (Table 6)

design, as well as for presenting the Mann–Whitney U-test *p*-values for individual vs. group testing. Restructuring our results for individual vs. group performance, we may state that the trends for group performance differ from those for individual one. As Tables 5 and 6 demonstrate, in three cases out of four (monotony/aesthetic, monotony/non-aesthetic, anxiety/aesthetic, anxiety/non-aesthetic), the group problem-solving enhances the cumulative impact of website aesthetics and task features upon user states. With non-aesthetic web design, collaborative activity leads to user dysfunctionality in combination with any type of tasks: With monotonous tasks, the group environment contributes to the development of fatigue, while with tasks that cause anxiety, anxiety increases, and, in both cases, lability decreases. Aesthetic design partly helps compensate for that in monotony-inducing work, but enhances dysfunctionality when tasks cause anxiety.

Performing the task individually may, in case of non-aesthetic design, help neutralize the impact of group task-solving, especially for monotony. With monotonous tasks, using better design and/or switching to individual modes of task solution may bring on better performance. With tasks that imply rapid shifts or intensification in time, though, perfect design causes a reverse effect in both group and individual performance; however, anyway, loss of intellectual lability in anxiety-inducing tasks performed individually is nearly two times smaller overall than in groups, and the rise of anxiety is also 127% (for non-efficient design) to 160% (for efficient design) for group performance as compared to individual. Thus, individual work upon both types of tasks is recommended.

*RQ3: What are the **cumulative effects** of the combination of **task features**, **web aesthetics**, and **testing circumstances**?*

As deduced from what is said above, it is evident that task features, in general, create two patterns of user reaction, just as they were meant to do. In monotonous tasks, monotony shows up via the pattern 'drop of intellectual lability + growth of fatigue', and for anxiety-inducing tasks the pattern is 'drop of intellectual lability + growth of anxiety.' In both cases, the other three emotional indicators remain stable. This hints towards possible independence of emotions within the users' emotional states and a possibility of reduction of particular emotions in work/study processes via focused changes in work tasks.

However, the combination of product features and individual vs. group regime mediates user performance multi-directionally. Thus, the results for intellectual lability are, in general, better for the individual regime and tasks associated with increased anxiety, whereas, for the monotony-related tasks, this cannot be said. The influence of aesthetic and non-aesthetic design is manifested differently here. The multi-dimensionality of the experiment allows for seeing the differences created by combination of the 'contextual fidelity model' factors not seen otherwise.

In particular, the cumulative negative impact of aesthetic web design and group activity is manifested only in combination with the anxiety-forming task. Here, the maximum decrease in intellectual lability ($-27.76$ (5.977)) and the maximum rise of anxiety ($|1.6|$ (0.699)) is observed. Thus, the worst combination of factors may be detected, which is 'anxiety-inducing task + aesthetic design + group testing.' For non-aesthetic design, this effect diminishes, however it is still evident, too. Individual testing further diminishes the effect, especially in terms of emotions, and, in particular, the combination of individual regime and non-aesthetic design helps lower the dysfunctional state significantly, as intellectual lability remains stable. Visual observation of the assessors even hints to some manifestations of hyperlability on non-aesthetic designs – the effect when anxiety rises and lability for multi-tasking remains the same. However, the anxiety effect of nervousness and lowered lability is there for the three other combinations of factors.

**Table 5.** The results of the experiment, aesthetic design, individual vs. group testing.

| User State | | Metrics | Aesthetic Design | | | | | | Mann–Whitney, Δ p-Value |
|---|---|---|---|---|---|---|---|---|---|
| | | | Individual Testing | | | Group Testing | | | |
| | | | before Task | after Task | Delta (Δ) | before Task | after Task | Delta (Δ) | |
| Monotony | | Intellectual lability (A$_{mean}$) | **187.05** **(1.057)** | **182.93** **(1.128)** | **−4.12** **(1.504)** | 186.9 (3.478) | 185.14 (8.230) | −1.76 (6.401) | 0.218 |
| | Emotional stress (10 to 1) | Calmness/anxiety | 7 (0.471) | 7 (0.480) | 0 (0.667) | 6.9 (1.286) | 6.8 (1.135) | −0.1 (0.316) | 0.796 |
| | | Energy/fatigue | **6.9** **(1.197)** | **5.5** **(0.527)** | −1.4 (1.505) | 5.9 (0.737) | 5.8 (0.788) | −0.1 (0.568) | **0.035** |
| | | Excitement/depression | 6.9 (0.567) | 6.9 (0.316) | 0 (0.817) | 7.2 (0.632) | 6.9 (0.567) | −0.3 (0.483) | 0.247 |
| | | Self-conf./helplessness | 6.85 (1.475) | 7.1 (1.100) | 0.3 (1.767) | 6.5 (1.080) | 6.8 (0.918) | 0.3 (1.059) | 0.796 |
| Anxiety | | Intellectual lability (A$_{mean}$) | **187.15** **(1.285)** | **171.13** **(3.067)** | **−16.02** **(2.791)** | **186.96** **(1.028)** | 159.2 (5.860) | −27.76 (5.977) | **0.000** |
| | Emotional stress (10 to 1) | Calmness/anxiety | **7.1** **(0.875)** | **6.1** **(0.800)** | **−1** **(0.471)** | 6.7 (1.330) | 5.1 (1.286) | −1.6 (0.699) | **0.000** |
| | | Energy/fatigue | 6 (0.670) | 5.9 (0.737) | −0.1 (0.568) | 5.9 (0.737) | 5.8 (0.788) | −0.1 (0.568) | 1.000 |
| | | Excitement/depression | 6.9 (0.567) | 7 (0.471) | 0.1 (0.567) | 6.9 (0.567) | 7 (0.471) | 0.1 (0.568) | 1.000 |
| | | Self-conf./helplessness | 7.1 (1.370) | 7.3 (1.251) | 0.2 (0.789) | 6.9 (0.500) | 7 (1.054) | 0.1 (0.738) | 0.796 |

*Note.* The values with standard deviations lower than deltas and *p*-values ≤ 0.05 are highlighted.

**Table 6.** The results of the experiment, non-aesthetic design, individual vs. group testing.

| User State | | Metrics | Non-Aesthetic Design | | | | | | Mann–Whitney, Δp-Value |
|---|---|---|---|---|---|---|---|---|---|
| | | | Individual Testing | | | Group Testing | | | |
| | | | before Task | after Task | Delta (Δ) | before Task | after Task | Delta (Δ) | |
| Monotony | | Intellectual lability (A$_{mean}$) | **186.74** **(1.266)** | **178.91** **(2.166)** | **−7.83** **(3.019)** | 186.92 (3.030) | 175.94 (3.815) | −10.98 (5.622) | 0.143 |
| | Emotional stress (10 to 1) | Calmness/anxiety | 7 (1.490) | 6.9 (1.449) | −0.1 (1.853) | 7.1 (0.875) | 6.9 (1.286) | −0.2 (0.919) | 0.971 |
| | | Energy/fatigue | 6.9 (1.449) | 5.5 (1.649) | −1.4 (2.547) | **6.9** **(1.197)** | **5.3** **(1.059)** | **−1.6** **(1.579)** | 0.912 |
| | | Excitement/depression | 7.1 (1.728) | 7 (1.247) | −0.1 (0.876) | 7.1 (1.286) | 6.8 (0.918) | −0.3 (0.949) | 0.853 |
| | | Self-conf./helplessness | 7.1 (1.370) | 7 (0.816) | −0.1 (0.876) | 6.7 (2.110) | 6.9 (2.020) | 0.2 (0.632) | 0.280 |
| Anxiety | | Intellectual lability (A$_{mean}$) | 186.94 (0.751) | 187.04 (4.247) | 0.1 (4.423) | **186.99** **(0.741)** | 173.94 (3.048) | −13.05 (3.497) | **0.000** |
| | Emotional stress (10 to 1) | Calmness/anxiety | **6.8** **(1.135)** | **5.7** **(0.823)** | **−1.1** **(0.568)** | 7.1 (0.875) | 5.7 (0.823) | −1.4 (0.966) | 0.631 |
| | | Energy/fatigue | 6.1 (0.567) | 5.9 (0.737) | −0.2 (0.632) | 6.1 (0.875) | 5.9 (1.449) | −0.2 (1.549) | 0.971 |
| | | Excitement/depression | 7.1 (0.994) | 7.2 (1.135) | 0.1 (0.568) | 6.9 (0.737) | 7 (0.816) | 0.1 (0.316) | 0.971 |
| | | Self-conf./helplessness | 7 (1.247) | 6.9 (0.875) | −0.1 (0.994) | 6.1 (1.663) | 5.9 (1.286) | −0.2 (0.422) | 0.853 |

*Note.* The values with standard deviations lower than deltas and *p*-values ≤ 0.05 are highlighted.

For monotonous tasks, the cumulative effects of contextual fidelity factors show up more for non-aesthetic design, which is somewhat more intuitively understandable. Thus, here, the worst combination is that of group work and non-aesthetic design, which is in line with expectations. Aesthetic design compensates the effect nearly completely in group testing where distractions brought by people surrounding an assessor create alternative foci of attention and raise the assessors' mood.

The cumulative effects discovered in our work may in short be described as follows:

**monotonous task + group work + non-aesthetic design**
monotonous task + individual work + aesthetic design　　　$\geq$ monotony effect
monotonous task + individual work + non-aesthetic design

**anxiety task + group work + aesthetic design**
anxiety task + group work + non-aesthetic design　　$\geq$ anxiety effect
anxiety task + individual work + aesthetic design

anxiety task + individual work + non-aesthetic design $\geq$ signs of hyperlability effect

Thus, the worst options are, for monotony: non-aesthetic design + group work, and for anxiety: aesthetic design + group work. The best combinations are, for monotony: aesthetic design + group work, and for anxiety: non-aesthetic design + individual work.

Given all stated above, we recommend individual task completion in real life if very good design is not guaranteed; however, the linkage of good design with task features still demands further investigation.

As noted in Section 4, we have also tested the hypotheses found in the extant literature: H1 on intensification of formation of dysfunctional tasks in group work, H2 on bad effects of group testing combined with good design for both types of tasks, and H3 on combination of non-aesthetic design and group work leading to growth of task-related emotional states and decrease of intellectual lability.

Our results show that, for both types of tasks, group work enhances the dysfunctional states associated with it. This allows us to claim the full manifestation of the cumulative effect for the combination of low-quality web design, tasks of both types, and the group testing environment. It also indicates the complete confirmation of **H3** and a partial confirmation of **H1**.

As already stated above, the dynamics of the formation of dysfunctional states is quite peculiar for high-quality design. We have found two effects that can be described as antagonistic. First, for monotonous tasks, the tendency of decrease in intellectual lability, in the individual test (~187 $\geq$ ~183) is slightly reduced and becomes non-significant in the group testing (~187 $\geq$ ~185). Similarly, the tendency of increase of fatigue true for the individual testing is not confirmed when a monotonous task is performed in a group. Thus, group testing actually reverses the negative effects of high-quality design noticed by the previous research [17], rather than fosters them. This rejects **H2** for the monotonous task while confirming it for the anxiety-inducing one where, with group testing, indeed, lability drops nearly twice and anxiety rises 160%. This allows us to point out that the findings by the authors [17] are confirmed for anxiety-inducing tasks only, and their results might have actually been conditioned by the experiment circumstances and the type of task. Moreover, aesthetic impact in the conditions of group testing when solving monotonous tasks demonstrates its compensatory character: It slows down the development of stressful conditions and helps maintain concentration.

The second antagonistic effect is also related to aesthetic design, but in its competition with non-aesthetic one. For anxiety-inducing tasks, aesthetic design provides for worse results than non-aesthetic one in any testing conditions. 'Bad' design evidently compensates for the nature of the task in terms of intellectual lability in both testing conditions and also slightly reduces anxiety in group testing. This needs more research, as stated above.

## 7. Conclusions

### 7.1. Further Discussion of the Results

Our work has shown that factors that affect user performance in solving web design-related tasks have to be assessed complexly, as individual assessment of single factors does not explain the effects discovered by earlier literature, or such effects may be conditioned

by more factors than described in literature. We have shown that co-assessment of the 'contextual fidelity model' factors demonstrates divergent patterns of their cumulative impact, where previously effects were considered one-dimensional. This approach may have greater explanatory power when effects do not correspond to expectations.

Extending our previous work on the contextual fidelity factors [18,22,28], we have linked it to the activity theory that provides for complex understanding of user reactions to multi-factor stimuli from the functional viewpoint, as it sees resulting user states as complex (intellectual + emotional) and (dys)functional, which actually describes the user states researched upon in DUXU but does not name them properly.

In our interpretation of the contextual fidelity factors, we have chosen the way opposite to the most research designs that orient to narrow and single proxies for each model factor. In contrast, we have used the U-index that includes 13 other parameters for product features, state-inducing rather than imitational tasks for task features, cultural differences as the major proxy for user traits difference, and individual/group performance as the test condition that casts most impact upon user performance. This strategy had been implemented in our previous work [28], but on much smaller data (16 vs. 80 assessors). For this paper, we have extended our sample and have also performed cross-cultural analysis, thus testing all the contextual fidelity factors in one research design; cultural differences, though, did not play any substantial role in how user perceptions worked and how the dysfunctional states formed. This gives a hint that dysfunctional user states in user experience may have a universal character not much dependent on cultural differences, but, of course, our data are not enough to tell it for sure.

We have detected two effects that have manifested within several combinations of factors, namely the monotony effect and the anxiety effect. While we, indeed, meant to induce these effects, we did not know which combinations of factors would lead to them, and which features of mental/emotional states would be responsible for them. We have shown that these effects may have cumulative nature—that is, they may grow when combinations of factors change. We have shown that they may also have antagonistic nature, going against expectations in terms of what factors compensate the negative cumulative effects. There were also signs of effects that need further investigation, namely, the hyperlability effect in the state of anxiety forming on low-quality web pages and the nearly-significant cultural differences in formation of excitement.

As to the impact of individual contextual fidelity factors within cumulative effects of dysfunctionality, we have shown that the testing environment plays a leading role in usability performance, as the results of the study confirm its influence on nearly any combination of other factors. Undoubtedly, other contextual fidelity factors also turn out to be significant for cognitive performance and emotional states of users. At the same time, the significance of these factors is unequal. Thus, in the proposed study, the next in importance after the individual/group format is the type of task. In particular, in the formation of anxiety, the cumulative effect is observed regardless of the aesthetic quality of the product. To the greatest extent, this effect affects intellectual lability. Anxiety that is implied by the nature of such tasks is the most destructive for the users' cognitive efficiency. The depth of decrease in intellectual lability for a forced task differs several times from the indicators for solving a monotonous task, both in individual and group testing settings.

Additionally, the lack of connection between cognitive and emotional stress indicators, on one hand, and the aesthetic quality of layouts, on the other, confirms the decisive importance of task features during tests, as their features may affect user performance via affecting particular elements of user states like mental lability, anxiety, or fatigue. The task features must not be taken for product features. This demands more complicated research designs where task complexity should vary, in order to delineate the impact of the web layout itself from the impact of task features.

Based on our results, we have recommended that real-world tasks are performed in individual regime rather than in groups, which may have important implications for the organization of workspace and schooling.

*7.2. The Limitations of the Study*

Our study has several limitations that deserve to be discussed.

1.  Our research design implies that the smallest-level sub-group has 5 assessors only. This is not much, despite the overall number of assessors being three to five times larger than in most DUXU studies on individual contextual fidelity factors. We have tried to overcome his limitation by enlarging the sub-groups for the first and the second stage of data analysis. In the research design, the quality dichotomies for each contextual fidelity factor are actually tested on 80 participants.

2.  Such research design implies that its general limitation is that metrics of validity and significance, including standard deviations and confidence intervals, may play smaller roles than for larger-sample studies. This general limitation is well-recognized as a basic limitation in the quantitative DUXU research; this is why we speak of effects but also call them trends or tendencies. Despite that, the results of studies that do not employ validity and/or significance testing are also recognized as important for user experience studies.

3.  The theory of dysfunctional states is new for the English-language DUXU studies, while, in psychology of work, there are important empirical studies based on it. Its unrelatedness to user experience studies may be seen as a limitation of our study, but we would insist on introducing it into the research area that highly depends of the rise of dysfunctionality and has actually tried a lot to detect the dysfunctional states without naming them and seeing them as complex user states.

4.  Two more limitations are linked to the method of scaling, via which emotional stress indicators were measured. First, the method implies associating one's state with a small enough number of fixed phrases, rather than more flexible measurement scales like, e.g., percentages or open questions that would allow self-interpretation of user states. Having a set of ranged phrases may lead to choices shifted towards either the middle of the scale or to some psychologically more acceptable options. Our results suggest that choices of the middle option occurred rarely enough, but the choice of the value slightly higher than the center was frequent, which deserves a separate methodological enquiry, despite the wide use of these questions-based Likert scales. Second, the questions could be understood differently in the two cultural groups. We have tried to overcome this difference by providing the questionnaires in English to both groups who we non-native English speakers, but this does not fully guarantee the equality of understanding.

5.  While working upon the data, we have discussed in the working group that, despite the instructions given to the assessors, their initial functional state, including the mood and mental readiness for task completion, is not fully captured by the tests before task and is elusive but potentially influential. In other words, factors beyond the immediate setting of the experiment may affect user performance in the ways not captured by the current testing instruments. This is why we have focused on the deltas that show the shifts in performance quality regardless of the underlying worse or better assessors' moods and their individual cognitive differences.

6.  In assessing web aesthetics, we have followed the logic of maximum divergence of attribute and, thus, have focused on web pages with either high or low U-index, but the pages with middle-range U-index values have remained untested. However, given the antagonistic effects discovered and the compensatory effects of non-aesthetic design in case of anxiety-inducing tasks, web pages of middle quality may happen to have optimal efficiency if both types of tasks are taken into account. This deserves further studies.

## 8. Conclusions: Questions and Prospects for Future Research

Our research has put forward several further questions. The first of them can be formulated as follows: Which of the 'contextual fidelity model' factors contribute most to the development of cumulative effects and the unfavorable functional states caused by

them? This question may seriously expand the emergent study area of cumulative effects in DUXU.

The difference in test results for the two types of tasks allows for a second question for further research—what could be the reason for the difference in the rate of decline in intellectual lability for a monotonous task? As can be seen from the results, in the case of a low aesthetic quality portal (low U-index rate), the group environment contributes to a deeper decrease in intellectual lability than an individual test does. However, for the high aesthetic quality (high U-index rate), the group environment slows down both the development of fatigue and the decrease in intellectual lability.

The higher the aesthetic quality, the stronger the compensatory effect of design on the developing state of monotony. As in the case of anxiety, the negative manifestations of monotony include a drop in the ability to concentrate and self-control, difficulties in updating working memory, and a general decrease in the efficiency of thinking. However, unlike anxiety, these phenomena are caused by fatigue that develop as a result of performing stereotypical actions without clear criteria for their effectiveness. Under such conditions, the speed of reactions slows down and the concentration of attention on the task decreases, although the level of emotional stress increases slightly. Perhaps this increases the sensitivity to distractions, in particular, to the presence of other people in the test environment. From this point of view, low-quality design is perceived by the user as a source of visual noise, contributing to switching attention to factors that are not related to the task at hand. This demands further research.

Third, the role of the ethno-cultural differences is not clear enough. In this study, the participants' ethnic and cultural belonging has not actively manifested. However, this may be due to the dominance of the overall 'Western' shape of the testing factors: Students from China are immersed in a European-style educational environment, they study and pass tests in Western languages, and the aesthetics and functional capabilities of the material being tested are both products of the Western visual culture. Perhaps when testing an audience of truly diverse cultural environments, the differences in cognitive and emotional assessments will be more noticeable.

Fourth, our study does not sufficiently illuminate the time factor, namely the duration of exposure to dysfunctional states. Any functional state can have three types of longitudinal impact: short-term, long-term, and chronic. This may demand that the activities or conditions necessary to compensate the dysfunctional states may need to be elaborated in different ways according to the longevity of effect. In this case, it is especially important to study the effects of dynamic mismatch: With a very small or very strong stress put upon the users' cognitive and emotional resources, the hypercompensation effect may occur, in which the user's psyche adapts to extreme operating modes. The effects of habituation and functional over-training may be caused by regular completion of tasks of a certain type in a high-intensity operation mode. Thus, in further studies, it would be useful to highlight the interaction effects of all testing factors under such peak loads, since user behavior in such situations (for example, searching for information under time pressure) can differ greatly from routine problem-solving.

**Author Contributions:** Conceptualization, A.V.Y. and S.S.B.; methodology, A.V.Y.; test conduct: A.V.Y.; data analysis: A.V.Y.; writing: A.V.Y. and S.S.B.; review and editing: S.S.B.; supervision: S.S.B.; project admin-istration, S.S.B.; funding acquisition, S.S.B. All authors have read and agreed to the published version of the manuscript.

**Funding:** This research was funded in full by the project 'Center for International Media Research' of St. Petersburg State University, year 2, grant #92564627.

**Acknowledgments:** The authors are grateful to the students of St. Petersburg universities who took part in the study.

**Conflicts of Interest:** The authors declare no conflict of interest.

## Appendix A. Examples of Web Pages Used for the Experiment

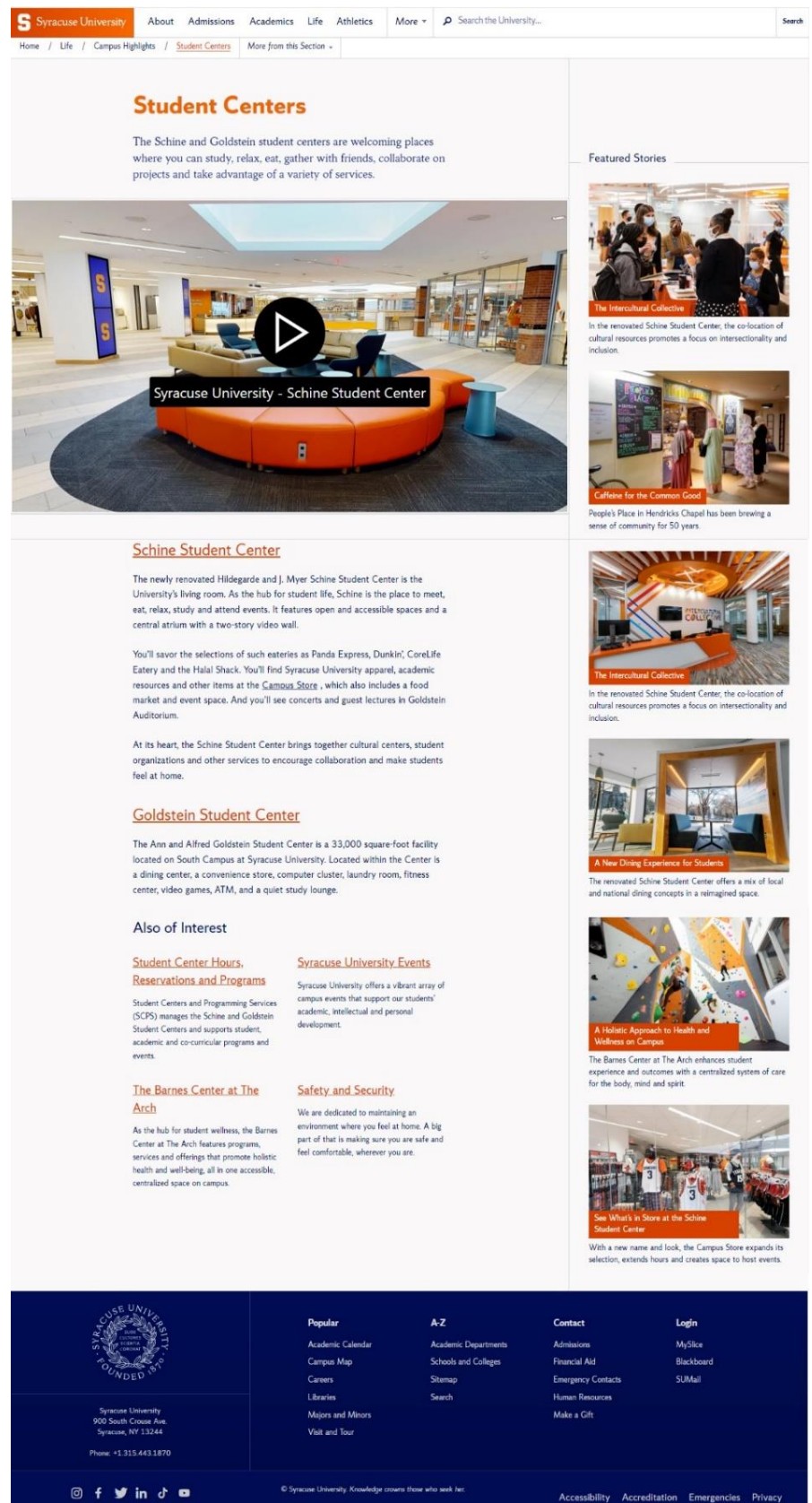

**Figure A1.** The web page of high aesthetic quality (Syracuse State University).

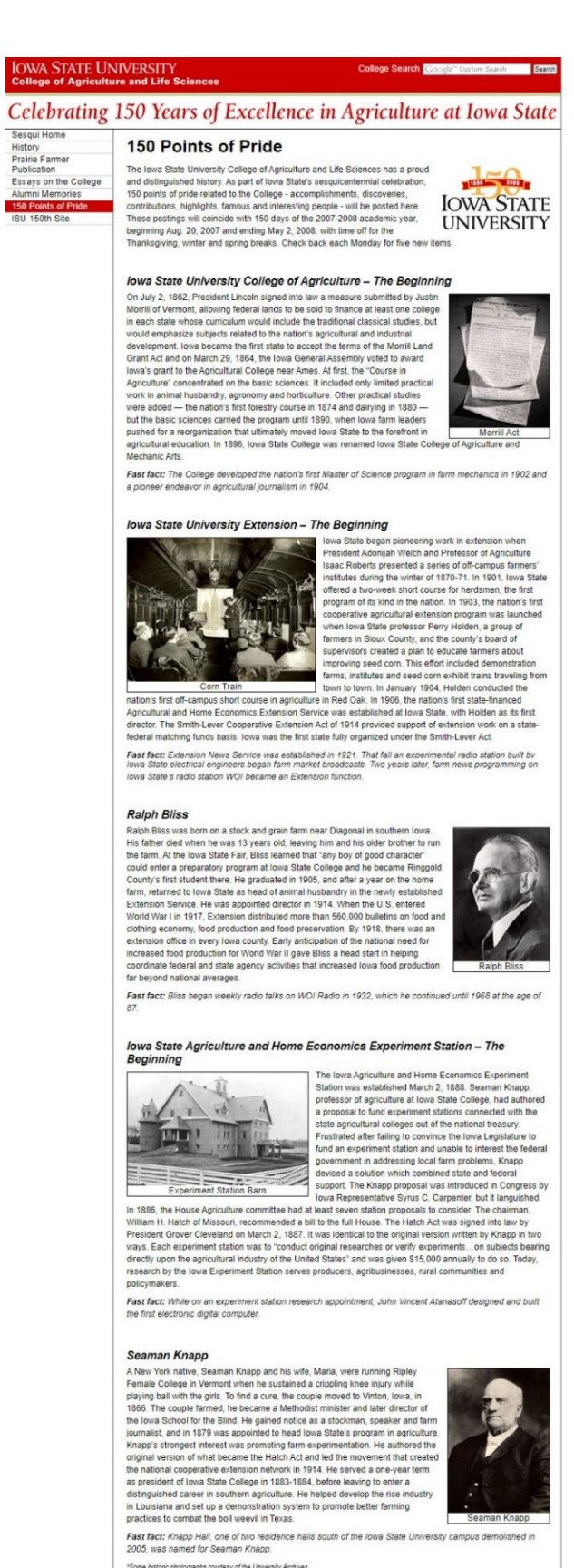

**Figure A2.** The web page of low aesthetic quality (Iowa State University of Science and Technology).

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
