# Peer review of "Cumulative Impact of Testing Factors in Usability Tests for Human-Centered Web Design†"

_futureinternet, doi:10.3390/fi14120359_

Round 1

Reviewer 1 Report

1. In section 3.1: It is not clear what does “Group participation” means. Performing the task as individuals is clear but as a group it’s not. Do assessors perform tasks all together in the same room? What difference does it make?

2. Assessor groups: There are 16 sub-groups. How many individuals form each group? How many students belong to group tests? Can the same student participate in different groups? I believe that this section needs revision so the reader can understand better the methodology.

3. Lines 44-50 lack of bibliographic references. In the line 46 (“…various characteristics within individual factors are investigated”) what are the characteristics that have been investigated? Also, in the line 47 “Such studies…” a reference that confirm this could be added.

4. I suggest the authors should expand on the Introduction. Although, it would be ideal to have three large paragraphs for instance, rather than many short ones. In general, the introduction could be more up - to - date.

5. The literature review is organized into sub-sections, which is good enough, however, I believe it should not be divided into so many short paragraphs. There may have also been more bibliographic references to strengthen the theoretical background.

6. Line 322: special characters “ *** “ should be removed.

7. Line 329: The research questions/hypothesis could be treated as a separate chapter rather than a sub-section as “2.6”.

8. Lines 428 and 445: Definitions for the two terms such “Intellectual liability” and “Emotional stress” may have been provided. 

9. Tables 2 and 3: which table is for East and which for West macro-groups? Both have the same caption.

10. Lack of a separate chapter “Discussion”.

11. Lack a separate chapter of “Limitations”.

Author Response

Dear Reviewer!

Thank you indeed for your attention and recommendations. Thanks to critique of yours and of the other reviewers, we have now significantly reworked the paper (please see green marks in the text and the responses to your remarks). We hope for your second-round review and are grateful in advance.

Our replies to your comments, one by one, are provided in the attached file.

Yours sincerely,

Authors

Reviewer 2 Report

Cumulative impact of testing factors in usability tests for human-centered web design

In the considered manuscript, the authors seek to study interaction (cumulative impact) of factors in usability testing. The positive aspects of the study include the potentially interesting experiment and the large sample of subjects from different countries, who took the time to perform tasks requiring significant effort. However, there are several major problems with the paper that do not allow me to recommend accepting it for publication in the Future Internet journal. My detailed comments and recommendations are below.

First, the paper lacks methodological focus. The authors try to cover a lot of different problems in the field, some of them quite general, but most of the things they mention do not make it into their actual experiment. So, the connection between the discussion of several high-level concepts (e.g., task complexity, aesthetic perception, activity theory, etc.) and the actual study often remains unclear.

In some cases, the connection is apparent, but the subsequent narrowfication makes the authors' initial claims doubtful. For instance, according to the Abstract, the study deals with 1) product properties, 2) task features, 3) user traits and 4) environmental/context factors. But for the experiment, the authors narrow down each of these to just one facet at best and actually deal with 1) website aesthetics, 2) individual/group test format, and 3) users' cultural group (as for the dysfunctional state variable in the experiment, I'm not sure if it can correspond to 4).

In general, it is not always clear how the methodological discussions relate to the actual experiment. For instance, "The resulting states of a user’s mind and mood, both during and after performance, need to be seen as complex and dynamic just as well" - but what should this mean in practice? What would be the difference if they are simple and static instead?
Another one: "The Russian psychological school has produced several important works in this respect, which we will mainly use as the basis of elaborating upon the target variables of our study." - but what variables and how exactly were they elaborated? In the actual experiment, intellectual lability was measured as per [37] and emotional stress as per [38] (the latter is Wessman, USA, 1966) - why the global reference to the "Russian psychological school"?

As a result, the conceptual validity of the study suffers - the experimental design does not correspond to the theoretical / research question part. Even if we consider "usability tests" (mentioned in the title): in practice, usability testing is usually an individual session lasting 60-90 minutes, with several tasks and the "think aloud" technique that is widely considered to be the key to the usability testing. Also, no usability testing has the goal of studying the users, as was the case in the authors' work. Meanwhile, there is no even discussion of these differences.

Second (related), the introduction and the state-of-the-art review are rather inordinate and outdated.
Compared to the actual work done, the theoretical part of the paper appears bloated, as the authors try to discuss too many things in the field. This could have been interesting to the reader, but the organization of the text also suffers and it's not always easy to follow the authors' thinking. For instance, in sub-section "2.1. User traits in usability testing studies", there are sentences like "The results of studies demonstrate both obvious patterns (high aesthetic quality of a product increases perceived usability) and paradoxical dependencies (high quality design contributes to lower productivity).", whose relation to user traits is unclear.
The notes that the authors make about the state-of-the-art might be reasonable, but most of them do not seem to relate to the current work. For example: "qualitative characteristics of tasks and their connection with the testing environment remain understudied" - but I do not see how the authors themselves study qualitative characteristics of tasks. I cannot agree that "a task causing anxiety" is really a task characteristic in the conventional sense.

The justification of the study novelty is not adequate either. Several times throughout the paper the authors claim that some problem or even field "remain heavily understudied", but they do not really reference or discuss up-to-date publications in the field. A significant part of the references in the list are from 10-15 years old, while the authors seem to treat them as recent. It makes me wonder if the paper's text was written quite a while ago.
For instance, "influence of aesthetic quality of a website upon user experience has become a growing area of research – which, however, remains quite narrow in its scope and highly under-researched" - the authors call it a "growing area", but most of the related references they cite are over 10 years old.
Another example: 209 "Recently, the research area dedicated to studying the cumulative nature of user experience [23]." - whereas the reference [23] is from 2011. I do not think that a publication from over 10 years ago is really a recent one.

So, I strongly recommend the authors to completely re-consider the theoretical part of their paper. Their work would benefit if the Introduction and the Methods are up-to-the-point (of the current study) and not too general.

Third, the statistical analysis is very poor.
Not much to detail here, just I have to say that virtually no statistical analysis is done. The one in "4.1. Testing the cultural differences" might be a reasonable one (though I am puzzled by so many p-values exactly equal to 1 in Table 2 and Table 3), but it yields no significance anyways.
Then, in 4.2, supposedly the main results of the study, there is just Table 4 with very basic descriptive statistics - the means without even deviations! No hypotheses testing is performed, but the authors claim that "It also indicates the complete confirmation of H3 and a partial confirmation of H1." - I have to say it is impossible to make such a conclusion by just looking at the means.   Thus, I suggest that the authors actually explore the effects and the interactions using appropriate statistical tests.

Finally, the conclusions are not supported by the results.
Correspondingly, most of the conclusions the authors make are not justified.
A most obvious problem is that the authors talk about significance, although no statistical tests were performed (besides the ones for the East-West factor, which was insignificant). E.g.: "Undoubtedly, other factors also turn out to be significant for the productivity and emotional states of users". This is not to mention that I don't quite understand what the authors mean by user productivity in their experiment.

Another example of making conclusions about the variables that do not seem to be considered is "The results of the study confirm its influence on any combination of other factors related to the nature of the task and product usability" - I don't see the product usability as a factor in the authors' experiment.

"We have shown that the testing format plays a leading role in usability testing" - yet another example of an improper generalization. Is "the testing format" limited to individual/group? "A leading role" among which factors? Also, I cannot agree that the experiment set up by the authors corresponds to conventional usability testing.   There is also no proper discussion of the limitations of the study. For example, if I got it right, the authors used just two websites (each for one task) - this is a very limited material, to say the least.

All in all, I recommend that the authors re-consider the Conclusions and Discussion section after obtaining the results of the analysis.

Misc:
DUXU is not a conventional abbreviation yet (especially for readers of Future Internet, which is not all about HCI) and must be clarified in the Abstract and in the text.

"We conclude by setting questions and prospects for further research." - I don't think this phrase belongs in the Abstract, I would rather recommend the authors to briefly outline the prospects.

328: "hupotheses"

"confirming the null hypothesis" - this can't be done in Statistics.

Author Response

Dear Reviewer!

Thank you indeed for your attention and recommendations. We have now significantly reworked the paper (please see the green marks in the new version of the paper). We hope for your second-round review and are grateful in advance.

Our replies to your comments, one by one, are provided in the attached file.

Yours sincerely,

Authors

Reviewer 3 Report

The present article describes the results of usability testing involving 80 individual assessors in both individual and group setting, on the basis of the “contextual fidelity model”. The innovative aspect of the study is that it focuses on establishing relationships between the model’s individual factors and testing their cumulative impact by assessing two user states of dysfunctionality: monotony and anxiety. 

The study sets the stage with a comprehensive literature review and proceeds to explain its methodology and results in detail. Some minor improvements that would improve the articles readability are as follows:

Line 348. You mention here “tasks of both types” without explicitly naming the types in this section of the article (I’m assuming monotony/anxiety inducing tasks based on section 3.2). Please very briefly clarify the types before posing the hypotheses.

Lines 448-451. The way the Likert scales are presented implies that higher value means  more fatigue or anxiety when this is not the case. Please try to more explicitly present what lower/higher values mean for each scale in the methodology. 

Section 4.2 In order to improve understanding on which cases confirm (or don’t confirm) the partially confirmed hypotheses H1 and H2, it might be useful to present them in a table. This table would be similar to table 4 but retaining only relevant metrics (intellectual liability and monotony for monotony inducing tasks and intellectual liability and anxiety for anxiety inducing tasks) and using natural language instead of numbers to indicate the confirmation or not of each hypothesis. Additionally, all three Hypotheses should be restated in this section to improve readability. 

Some minor proofreading remarks are:

Line 209. The sentence seems to lack a verb or a helping verb. Please restructure for clarification. 

Line 322.The text “***” seems to be a place holder for something? Was something omitted? 

Line 328. Hupotheses should be changed to Hypotheses.

Overall, the English used in this paper is good. Some minor editing and proofreading might be necessary. 

Author Response

Dear Reviewer!

Thank you indeed for your attention and recommendations, as well as for your favorable opinion on our paper. Thanks to comments received from you and other reviewers, we have now significantly reworked the paper (please see the green marks in the new version of the paper). We hope for your second-round review and are grateful in advance.

Our replies to your comments, one by one, are provided in the attached file.

Yours sincerely,

Authors 

Round 2

Reviewer 1 Report

I am satisfied by authors' reply but I do not believe that the authors responded satisfactorily to all other reviewers' comments.

Author Response

Dear Reviewer,

please see our response to Reviewer 2 attached to the system; it has taken us some time to answer in with proper rigor. We hope our answer will satisfy you. Thank you for not posing us questions this time :)

Yours sincerely,

authors

Reviewer 2 Report

I have read the other reviews and the authors' response. I commend the authors for making some improvements to the paper: in particular, adding and updating references and detailing the research design.

However, I was sorry to see the conceptual validity of the study has not been improved considerably. The authors insist on over-generalizations, although I do not see why the things cannot be called by their precise names.
"product features -> web design quality -> U-index (13 parameters)" - this might be right, but the arrows do not work in the other direction. U-index cannot stand for web design quality (there are other aspects and measurements). Web design quality can not stand for product features (there are also e.g. functional features).
"user productivity (as short for ‘user’s productivity of cognitive operations’)" - user productivity of cognitive operations (whatever this is) cannot stand for user productivity in general. Using a more general term instead of a more specific term (that is actually employed in a study) is misleading and must be avoided.
"We have shown that the testing format plays a leading role in usability testing" used in the paper is not the same as "group testing format affects the users’ results in more cases than other contextual fidelity factors", which the authors use in their response. Over-generalizations must be avoided.
"Product features clearly affect the results..." - whereas my initial comment was about product usability, not about product features. It referred to the sentence "The results of the study confirm its influence on any combination of other factors related to the nature of the task and product usability" that is still persistent in the current version of the manuscript. I insist that the product usability was not considered in the experiment.
The end result of the overall conceptual neglectance is "three hypotheses (not directly stemming from the RQs)", which kind of contradicts the idea of RQs and hypotheses.

Some conceptual statements made by the authors are plain wrong.
For instance, "Independent variables were formulated according to the ‘contextual fidelity’ model which is an academically recognized model of effects in usability testing.":
a query to Google Scholar
https://scholar.google.com/scholar?hl=en&as_sdt=0%2C5&q=%22contextual+fidelity+model%22+in+%22usability+testing%22&btnG=
reveals that contextual fidelity model in combination with usability testing is used only by three people in the world, two of whom are the authors of the current manuscript.
I shall repeat that Usability Testing is a qualitative method, which has nothing to do with independent variables or effects.
Moreover, it has nothing to do with studying the users, which belongs to User Research (see e.g. https://en.wikipedia.org/wiki/User_research ), particularly since Usability Testing is usually done with groups of 3-5 participants.
"the ‘think aloud’ technique is used even more rarely than the Likert scales of various sort for subjective assessment of user experience" - indeed, because it is used to assess the user interface(s), which is the goal of Usability Testing.
I strongly suggest that the authors familiarize themselves with various types of methods existing in HCI and use the right names.

As for the authors' response to my comment about the poor statistical analysis:
The sample size (80 participants) employed in the study is not problematic and is rather typical for experiments in the HCI field. However, the authors' claim that a dataset obtained with few subjects can not be analyzed using statistical methods is ridiculous. Particularly since they use this claim apparently to defend not using any statistical methods at all (and refer to CHI in this), but making conclusions about hypotheses nevertheless.
All in all, though the authors' approach to statistical testing in their work remains controversial and poorly explained, the reader now has the ability to judge about its validity, since more values are reported. I however insist that the authors must present the two websites used in their study and discuss this limitation. This is a very important threat to validity, as just two items of material surely have many more differentiating factors, unconsidered by the authors.
Misc:
189: "called the micro- and micro-" - probably one of them got to be "macro-"
in P. 13-16 the authors use decimal comma instead of dot

Author Response

Dear Reviewer,

Please see our answers to your queries attached to this letter. Thank you indeed for your attention to our paper, we are sincerely grateful for the amount of effort you have invested in making our paper better.

Yours,

authors

Round 3

Reviewer 2 Report

I have read the authors' reply to my comments and the revised version of the manuscript. I commend the authors for having added the screenshots of the websites (the material used in the experiment) and for clarifying and fixing some of the conceptual issues.

There seems to be some misunderstanding regarding my comments related to the conceptual part of the paper and the terminology. In their reply, the authors highlight that they "cannot agree that our choice [of proxies] is bad" and provide a justification for them. However, I have never questioned the choice of variables in the study. My recommendation was just to name them correctly, instead of using more general terms (which supposedly adds significance to the study, but ruins conceptual validity) or "shorter versions" (the example with the UK, which is a convention, not a scientific term, is hardly inappropriate). Cf. e.g. "density" and "information density".

The authors' reply about usability testing being a part of user research that  can rely on qualitative methods seems to be in line with the same (misleading) logic. My initial comment was exactly that that usability testing and user research are different things, not that they are not related.

UT = testing products ("Usability testing is a technique used to evaluate a product. This is done by testing it on users.")

UR = studying users + studying interaction + studying products in use + ...

hence

UR = UT + something else

not that

UT = studying users

In any case, in my previous review I have already recommended acceptance of the manuscript for Future Internet. So, I do not require any more improvements to the paper, and sincerely with the authors best luck in their future research.